# Responsive Acrylamide-Based Hydrogels: Advances in Interpenetrating Polymer Structures

**DOI:** 10.3390/gels10070414

**Published:** 2024-06-21

**Authors:** Lenka Hanyková, Julie Šťastná, Ivan Krakovský

**Affiliations:** Department of Macromolecular Physics, Faculty of Mathematics and Physics, Charles University, V Holešovičkách 2, 180 00 Prague, Czech Republic; chamky@seznam.cz (J.Š.); ivank@kmf.troja.mff.cuni.cz (I.K.)

**Keywords:** polymer hydrogel, hydrophilic polymer, acrylamide-based polymers, lower critical solution temperature, poly(*N*-isopropylacrylamide), poly(*N,N*-diethylacrylamide), interpenetrating polymer network, double network, stimuli-responsive polymers, temperature responsiveness, pH responsiveness, mechanical properties, drug delivery, biomedical applications

## Abstract

Hydrogels, composed of hydrophilic homopolymer or copolymer networks, have structures similar to natural living tissues, making them ideal for applications in drug delivery, tissue engineering, and biosensors. Since Wichterle and Lim first synthesized hydrogels in 1960, extensive research has led to various types with unique features. Responsive hydrogels, which undergo reversible structural changes when exposed to stimuli like temperature, pH, or specific molecules, are particularly promising. Temperature-sensitive hydrogels, which mimic biological processes, are the most studied, with poly(*N*-isopropylacrylamide) (PNIPAm) being prominent due to its lower critical solution temperature of around 32 °C. Additionally, pH-responsive hydrogels, composed of polyelectrolytes, change their structure in response to pH variations. Despite their potential, conventional hydrogels often lack mechanical strength. The double-network (DN) hydrogel approach, introduced by Gong in 2003, significantly enhanced mechanical properties, leading to innovations like shape-deformable DN hydrogels, organic/inorganic composites, and flexible display devices. These advancements highlight the potential of hydrogels in diverse fields requiring precise and adaptable material performance. In this review, we focus on advancements in the field of responsive acrylamide-based hydrogels with IPN structures, emphasizing the recent research on DN hydrogels.

## 1. Introduction

Hydrogels, which are three-dimensional polymeric networks capable of retaining substantial water content, consist of hydrophilic homopolymer or copolymer networks. Their porous and hydrophilic structures closely mimic those of natural living tissues, rendering them highly suitable for diverse applications such as drug delivery, tissue engineering, and biosensors [1,2,3,4]. The synthesis and structure of hydrogels were first reported by Wichterle and Lim in 1960, marking the inception of hydrogel research [5]. Since then, scientists have extensively explored this field, developing numerous types of hydrogels with distinct features and functionalities.

Responsive hydrogels, a subtype of hydrogels, exhibit reversible alterations in their structure or characteristics when exposed to external stimuli, representing a particularly promising area of research. These stimuli include variations in temperature, pH, light, or the presence of specific molecules. Such responsiveness enables these hydrogels to adapt dynamically to changing conditions, rendering them indispensable in fields requiring meticulous regulation of material performance [6,7,8]. Temperature is a crucial external stimulus, as many biological processes, such as enzyme activity, cell metabolism, and tissue regeneration, are temperature dependent. Temperature responsiveness thus stands out as the most extensively researched category among hydrogel network systems.

Amphiphilic polymers in aqueous solutions exhibit a temperature-dependent behavior known as the lower critical solution temperature (LCST) phenomenon. Below this threshold, these polymers dissolve readily, adopting a coil-like structure. However, once the temperature surpasses the LCST, they become immiscible, transitioning into a compact globular conformation [9,10,11]. This reversible transformation hinges on a shift in the balance of interactions, particularly between hydrogen bonds and hydrophobic interactions. This behavior parallels the volume phase transition observed in cross-linked hydrogels. Below their volume phase transition temperature (VPTT), hydrogels swell as they absorb water, while above it, they release water and shrink [12]. Both the phase separation in solutions and the volume phase transition in hydrogels are believed to originate from a coil-to-globule transition on a molecular level, detectable through light scattering techniques [13].

Among temperature-sensitive polymers, acrylamide-based polymers, particularly poly(*N*-isopropylacrylamide) (PNIPAm), have received the most extensive research attention. PNIPAm exhibits an LCST of approximately 32 °C, rendering it highly relevant for physiological applications requiring precise thermo-responsive behavior. The phase separation of linear PNIPAm and the transition of PNIPAm hydrogels have garnered considerable attention in scientific exploration [9,10,11]. Poly(*N*-isopropylmethacrylamide) (PNIPMAm), like PNIPAm, falls within the category of thermosensitive polymers. However, it contains a side methyl group on each monomer unit, distinguishing it from PNIPAm. This additional α-methyl group raises the LCST to 42 °C [13,14,15,16]. Poly(*N,N*-diethylacrylamide) (PDEAm) is another temperature-sensitive polymer with promising biocompatibility [17], which possesses an LCST within the range of 31–34 °C [18,19,20]. 

pH-responsive hydrogels have garnered significant attention in research. Typically comprising polyelectrolytes, these hydrogel networks feature chains abundant in ionizable or protonated groups. Alterations in pH induce electrostatic repulsions among these groups, thereby influencing hydrogen bonding interactions between inter-polymer chains and the ionization or protonation dynamics between polymer chains and water molecules [21]. Glucose-responsive hydrogels can specifically detect glucose in bodily fluids by utilizing glucose-sensitive components embedded within their structure. These hydrogels can then produce optical or electrochemical signals in response to changes in glucose concentration [22].

Conventional hydrogels face limitations in their mechanical characteristics, prompting a demand for enhanced mechanical performance across various applications. The double-network (DN) hydrogel approach, as outlined in [23], has demonstrated efficacy in bolstering both mechanical strength and fracture toughness. DN hydrogels form a unique class of interpenetrating network hydrogels, defined by two networks with distinctly asymmetric and contrasting properties. These networks vary in aspects such as cross-linking density, rigidity, and molecular weight. The first network is typically highly cross-linked and brittle, and after its formation, it is swollen in the monomer solution of the second network, which is then polymerized to form a loosely cross-linked, ductile network. The exceptional mechanical properties of DN hydrogels are a result of this distinctive dual-network structure. Building upon the DN concept, various innovative systems and techniques have been devised to enhance mechanical properties. These include DN hydrogels capable of controlled shape deformation [24], organic/inorganic DN hydrogel composites [25], and flexible display devices utilizing DN hydrogels [26]. 

This review covers sensitive acrylamide-based polymers, highlighting both the well-known PNIPAm and other acrylamide-based polymers that are less frequently discussed in the literature. We introduce hydrogels with IPN structures and systematically describe the advancements in responsive acrylamide-based IPN hydrogels. Special attention is given to DN hydrogels, emphasizing their exceptional mechanical properties and providing a comprehensive overview of the research on responsive acrylamide-based DN hydrogels.

## 2. Acrylamide-Based Responsive Polymers

### 2.1. Polyacrylamide

Polyacrylamide (PAAm, Figure 1a) hydrogels are networks of hydrophilic polymer chains that can absorb a significant amount of water, swelling to many times their original volume, and this swelling behavior is highly sensitive to environmental conditions. PAAm holds a prominent position among responsive polymers, as experimentally, first-order phase transitions have been demonstrated in hydrolyzed PAAm hydrogels immersed in water/acetone mixtures [12]. This phenomenon involves a dramatic change in the volume of polyacrylamide hydrogels when exposed to a water–acetone mixed solvent containing 40% acetone by volume. The underlying physics of this step change can be associated with the balance between the osmotic pressure driving the swelling and the polymer–solvent interactions. Besides a drastic change in swelling, this phenomenon is also accompanied by changes in other macroscopic properties, such as transparency, opacity, and mechanical properties (Figure 2a,b) [27]. Furthermore, for the neutral chains of the hydrogels, the changes in swelling are gradual, whereas for the charged chains, there is a jump in swelling dependencies (Figure 2c).

### 2.2. Poly(N-isopropylacrylamide)

Poly(*N*-isopropylacrylamide) (PNIPAm) has garnered significant attention over the past decades due to its thermo-responsive behavior within a temperature range of biomedical interest [28]. PNIPAm is a synthetic thermo-responsive polymer derived from the acrylamide monomer N-isopropylacrylamide (NIPAm), consisting of amide and isopropyl groups (Figure 1b). The first detailed study of aqueous PNIPAm solutions in relation to temperature was conducted by Heskins and Guillet in 1969 [29]. They visually observed changes in the clarity of the solution upon heating and determined that macroscopic phase separation of PNIPAm solutions occurs at 32 °C.

The thermal behavior of PNIPAm in aqueous media involves a phase transition from a hydrophilic to a hydrophobic state when heated above its LCST. Below the LCST, the hydrogel absorbs water and swells, but once the temperature exceeds the LCST, the PNIPAm network collapses and precipitates [9,10,11]. In the hydrated state below the LCST, water molecules form hydrogen bonds with the carbonyl and the nitrogen atom of the amide group (Figure 3a) [30]. Above the LCST, intramolecular hydrogen bonds rearrange, reducing the number of hydrogen bonds between PNIPAm and water, leading to the formation of intra-chain hydrogen bonds (Figure 3b).

PNIPAm-based hydrogels have attracted significant research interest due to their remarkable properties and functionalities. The volume phase transition temperature and the degree of swelling/shrinking in these hydrogels can be finely tuned through copolymerization with hydrophilic or hydrophobic monomers [31,32]. Notably, changes in the properties of PNIPAm-based hydrogels can be induced through various methods, including heating [33,34], photoionization [35], or photoisomerization [36]. This allows the hydrogels to respond to temperature, light, electrical fields, and magnetic fields. These exceptional properties enable PNIPAM-based hydrogels to perform diverse smart functions, such as shape transformation, self-regulation, and rupture, akin to the behavior of cells and certain intelligent biological systems.

From a physiological perspective, the swelling transition of PNIPAm-based hydrogels at around 32 °C, close to body temperature, makes them highly attractive for various biomedical applications, such as drug delivery systems [37,38] or cell scaffolds in tissue engineering [39,40]. Due to the switchable optical properties that align with the thermo-responsive characteristics of PNIPAm-based hydrogels (such as temperature-induced changes in volume and transparency), researchers have extensively focused on developing PNIPAm-based hydrogels for use in smart optical materials across various fields as photonic crystals or smart windows [41,42]. PNIPAm-based hydrogels can undergo macroscopic motions and deformations due to temperature-induced volume changes, leading to the development of versatile hydrogel actuators [43,44].

### 2.3. Poly(N-isopropylmethacrylamide)

Poly(*N*-isopropylmethacrylamide) (PNIPMAm) closely resembles PNIPAm, differing only by an extra methyl group on the backbone (Figure 1c). Like PNIPAm, PNIPMAm exhibits LCST behavior in aqueous solutions. However, its coil-to-globule transition occurs at a higher temperature of approximately 44 °C, and this transition temperature is largely unaffected by molar mass, concentration, and the presence of added electrolytes [11,45]. The exact reason for the higher LCST of PNIPMAm compared to PNIPAm is still debated, but it likely involves the steric effect of the additional methyl group influencing chain conformation and hydration. Moreover, PNIPMAm chains are more hydrated and expanded below LCST and form more loosely packed aggregates above critical temperature compared to PNIPAm [13,15,46,47,48].

Hydrogels synthesized solely from PNIPMAm are not frequently studied. Instead, research typically focuses on hydrogels copolymerized with other components or incorporated into IPNs to tune the VPTT, such as in polyurethane/PNIPAm/PNIPMAm hydrogels [49]. The study of the volume phase transition in PNIPMAm/PNIPAm IPN hydrogels with varying PNIPMAm content revealed that only the hydrogel with ~54% PNIPAm exhibited a two-step collapse. In contrast, samples with higher PNIPAm content showed a single transition, indicating enhanced mutual entanglement of IPN chains [50]. The preparation and characterization of P(NIPAm-co-NIPMAm)/chitosan core/shell nanohydrogels, capable of drug release at different temperatures and pH levels, were reported [51]. Recently, it was shown that DNA modification reduces the volume phase transition temperature of PNIPMAm microgels, which could be further lowered by salt [52].

### 2.4. Poly(N,N-diethylacrylamide)

Poly(*N,N*-diethylacrylamide) (PDEAm), with an LCST of approximately 31 °C, is an attractive alternative to PNIPAm due to its better biocompatibility and similar structure [53] (Figure 1d). Unlike PNIPAm, which has a secondary amide group that can act as both a hydrogen bond donor and acceptor, PDEAm has a tertiary amide group that can only accept hydrogen bonds and cannot form intra- or interchain hydrogen bonds [54]. Due to this structural difference, PDEAm is often compared to PNIPAm. Studies have shown that the swelling ratio of PDEAm hydrogel below its LCST is much lower than that of PNIPAm hydrogel with a similar monomer/cross-linker ratio, attributed to the differing chemical structures of their monomeric side chains [55].

Due to the better biocompatibility, the responsive behavior and biocompatibility of PDEAm, along with its derivatives and hydrogels, have garnered increased attention in recent years. The development of PDEAm hydrogels closely follows that of PNIPAm hydrogels. Consequently, studies on dual temperature- and pH-sensitive SIPN PDEAm hydrogels, incorporating poly(diallyldimethylammonium chloride) or poly((2-dimethylamino)ethyl methacrylate) as a second component, can be found [56,57]. Furthermore, *N,N*-dimethylacrylamide and *N*-vinyl-2-pyrrolidone effectively modified the thermo-responsive properties of DEAm-based hydrogels [58]. These hydrogels exhibited reversible swelling and deswelling behavior in buffer media when subjected to temperature changes across their critical temperatures. The potential of salecan/PDEAm hydrogels as anionic drug delivery matrices, using diclofenac sodium as a model drug, was investigated [59]. The shape and size of gold nanoparticles adsorbed on PDEAm microgels were shown to be regulated by the swelling degree and cross-linking density [60]. Poly(vinyl alcohol)(PVA)-graft-PDEAm copolymer and its membranes displayed thermo-responsive behavior in water and higher swelling ratio than PVA hydrogel and in vitro tests confirmed their biocompatibility [61].

## 3. Interpenetrating Polymer Network Hydrogels 

Interpenetrating polymer networks (IPNs) are a type of polymer material composed of two or more polymer networks that are physically intertwined on a molecular scale but not covalently bonded to each other [62]. These networks are interlaced but maintain their distinct chemical identities. The unique structure of IPNs provides a combination of properties from the constituent polymers, often resulting in enhanced mechanical, thermal, and chemical characteristics. Semi-IPNs (SIPNs) are crafted similarly to IPNs, where only one of the polymer networks is cross-linked, while the other remains a linear or branched polymer. From a synthetic perspective, IPNs could have two different preparation sequences during the preparation process: (i) simultaneous IPNs, where both polymer networks are synthesized simultaneously through polymerization processes that occur at the same time, or (ii) sequential IPNs, where one polymer network is synthesized first, and then, the second network is formed within the pre-existing network. This process typically involves immersing single-network hydrogels into a solution that contains a mixture of monomers, initiators, and activators, optionally including a cross-linking agent (Figure 4) [63].

### Acrylamide-Based IPN Responsive Hydrogels

The exceptional stretchability of IPN hydrogels is essential for preserving their original shape after extensive deformation. These hydrogels consist of intertwined long-chain polymer networks, cross-linked to form a unique structure. The entanglement and physical interactions among diverse polymers create a stable configuration, allowing them to maintain integrity after deformation, resulting in better mechanical properties than single-component hydrogels [64,65]. Beyond enhanced mechanical properties, the key feature of stimuli-responsive “smart” IPN hydrogels is their sensitivity to external physical and chemical stimuli. Their unique interpenetrating structure allows for the integration of networks with diverse properties during gel design and preparation, endowing them with a wide range of rich characteristics.

The IPN strategy enables a significant improvement of the essential properties of temperature-sensitive hydrogels prepared from PNIPAm. 

When the temperature rises above the VPTT, hydrophobic interactions between polymer chains occur, causing the collapse of the 3D hydrogel structure and expelling almost all the contained water. However, a dense layer forms on the hydrogel’s surface due to phase separation. This layer significantly hinders the outward diffusion of water molecules. As a result, the contraction kinetics of typical PNIPAm hydrogels are relatively slow, limiting their usefulness in fields requiring rapid response times [66,67]. 

Moreover, PNIPAm hydrogels have weak mechanical properties due to their low polymer density in the swollen state, reducing their thermal responsiveness and potential applications [68]. To address these issues, researchers suggest adding a second polymer to create an IPN structure. This method adjusts the phase transition temperature to body temperature and increases polymer density, enhancing mechanical properties. 

During the creation of IPN networks, channels are formed that improve water diffusion, accelerating the temperature response [69]. Zhang et al. integrated PVA into the PNIPAm network, creating an SIPN hydrogel [70]. PVA chains serve as water release channels, breaking the dense surface layer and significantly speeding up the response. This thermo-responsive IPN system extends use of acrylamide-based polymers in injectable hydrogels, drug delivery, release control, and cell delivery [71,72]. Ge et al. introduced a novel IPN hydrogel, combining PNIPAm and the upper critical solution temperature thermo-responsive poly(N-acryloylglycine) (PNAGA). This hydrogel transitions between swollen and deswollen states over a tunable temperature range of 0 °C to 65.8 °C. The structure and hydrogen bonds between the polymers enhance mechanical strength and self-healing, making it ideal for electronic skin, wearable tech, and bionics [73].

A two-layer design for stimuli-responsive hydrogels to introduce anisotropy into polymer structures is often used. This design consists of an inert supportive layer and a swellable stimuli-responsive layer, leading to uneven anisotropic expansion and shape changes upon external stimulus. For example, Liu et al. created a bilayer structure with a IPN PNIPAm/PAAm hydrogel on a PAAm substrate. This bilayer structure demonstrates reversible opening and closing actions under NIR illumination, similar to the human iris’s response to light changes [74].

The IPN approach enables the synthesis of a variety of multi-responsive hydrogels by combining single-network hydrogels with different stimulus responses. Temperature and pH are key triggers for IPN hydrogels, making them a focus of research. PNIPAm often forms thermo-sensitive networks in multi-responsive IPNs [9,75]. It combines with anionic pH-responsive polymers like polyacrylic acid [76], polyaspartic acid [77], or hyaluronic acid [78], exhibiting significant swelling transitions below the LCST and at higher pH levels.

A glucose-responsive IPN hydrogel is a composite material engineered by incorporating glucose-sensitive components into polymer network. These hydrogels can selectively interact with glucose, allowing for precise detection and response to glucose levels, making them valuable for applications in biosensors, drug delivery systems, and diabetes management. The glucose-responsive “smart” IPN hydrogels overcome issues like poor mechanical compliance and limited drug release control in conventional hydrogels. Ye et al. developed such a ConA@PNIPAm system incorporating concanavalin A (ConA) to create a nano-IPN hydrogel that encapsulates insulin and adjusts its release rate in response to glucose fluctuations [79].

An overview of key publications contributing to the research on IPN acrylamide-based responsive hydrogels is given in Table 1.

## 4. Double-Network Hydrogels

In addition to exceptional stretchability, efficient energy dissipation is essential for hydrogels to achieve high fracture energy and toughness. Since the pioneering work of Gong et al. in 2003 on double-network (DN) hydrogels, the cleavage of polymer chains has become a common method for dissipating mechanical energy [23,98,99,100]. DN hydrogels constitute a distinct category of interpenetrating network hydrogels characterized by two networks with asymmetric and contrasting properties. These networks differ in parameters such as cross-linking density, rigidity, and molecular weight. The superior mechanical properties of DN hydrogels arise from the unique asymmetry of their dual-network structure. When subjected to deformation, internal fractures occur within the first network, effectively serving as supplementary cross-linkers that fortify the hydrogel [98,99,100].

DN hydrogels are typically synthesized using a two-step polymerization method first developed by Gong for poly (sodium 2-acrylamido-2-methyl-propanesulfonate) (PAMPS)/PAAm hydrogels [23]. Initially, a covalently cross-linked, rigid, and brittle first network is formed using strong polyelectrolytes. This hydrogel is then immersed in a solution with neutral second-network monomers, initiators, and cross-linkers. The high swelling capacity of the first network allows the second-network reactants to diffuse in, causing significant volume expansion. The second polymerization then creates a loosely cross-linked, soft, and ductile second network within the first. This method, which requires a large excess of second-network reactants, is widely used to prepare DN hydrogels with both networks chemically cross-linked.

Tough DN hydrogels were achieved with a tightly cross-linked first network and a loosely cross-linked second network. Furthermore, a high molar ratio of the second network polymer to the first network was necessary to ensure the second network dominated the DN system [23,101]. These parameters are key to understanding DN hydrogel damage. Load–unload tests showed significant hysteresis beyond a certain strain, indicating energy dissipation [102]. Research found that increasing the first network’s cross-linker concentration from 2% to 4% significantly raised the number of energy-dissipating backbone bonds, correlating with the overall toughness of DN [103]. Higher cross-linker concentrations led to shorter, more uniform strand lengths in the first network, contributing to fracture toughness. This finding highlights role the first network’s as a “sacrificial bond” for energy dissipation, enhancing the mechanical toughness of DN hydrogels. Using a probability distribution model, it was shown that higher cross-linker concentrations in the first network result in a narrower distribution of shorter strands. The breakage of these short strands significantly contributes to the DN system’s fracture toughness [104]. Figure 5 illustrates how the first densely cross-linked network absorbs energy by stretching and fracturing, stabilized by the second loosely cross-linked network, preventing DN system failure [105].

### 4.1. Temperature-Responsive Acrylamide-Based DN Hydrogels

PNIPAm typically serves as the key temperature-responsive element within DN hydrogels sensitive to temperature changes. Early research investigations into this area included studies on nanocomposite DN PNIPAm hydrogels, which involved the incorporation of polysiloxane nanoparticles [106]. Without nanoparticles, a DN hydrogel essentially retained its equilibrium swelling and reswelling kinetics but showed enhanced deswelling rates and extents, along with increased stiffness and strength. Introducing polysiloxane nanoparticles into DN hydrogels further modified these characteristics, with effects dependent on nanoparticle size and whether they were introduced during the formation of the first or second network. The ongoing aspect of this research aimed to merge DN design with micropatterning techniques to improve cell detachment from PNIPAm hydrogels through controlled deswelling-induced release [107]. The characteristics of DN membranes composed of PNIPAm and the electrostatic comonomer 2-acrylamido-2-methylpropane sulfonic acid (AMPS) were assessed to investigate their viability as self-cleaning membranes for implanted glucose biosensors [108].

Ultra-tough hydrogels with a compressive strength of ≈23 MPa were prepared using a novel combination of components [109]. The first network consisted of a tightly cross-linked, highly negatively charged PAMPS, while the second network was a temperature-sensitive PNIPAm copolymerized with zwitterionic [2-(methacryloyloxy)ethyl]dimethyl(3-sulfopropyl)ammonium hydroxide (MEDSAH). The unique mechanical properties of these hydrogels stem from the zwitterionic nature of the second network, which introduces additional electrostatic attractive forces between the anionic sulfonate groups of the PAMPS network and the cationic ammonium groups of the PNIPAm-co-MEDSAH network. Additionally, the relatively low content of MEDSAH copolymerized with NIPAm in the second network maintains the “PNIPAm-like” volume phase transition temperature (VPTT) at around 35 °C, making it suitable for biomedical applications.

PNIPAm was also utilized as the second component in the preparation of DN hydrogels designed for cooling applications [110]. These hydrogels consist of a superhydrophilic PAMPS network and a temperature-responsive PNIPAm network. The temperature-responsive network enables the hydrogel to achieve significant evaporative cooling at relatively low temperatures. Due to the presence of the superhydrophilic network, the rate of water release from the DN hydrogel is slower compared to that of a pure temperature-responsive, single-network hydrogel, resulting in extended cooling duration.

The temperature response of DN hydrogels made of thermo-responsive PNIPAm and hydrophilic PAAm was investigated in [111]. By combining swelling, mechanical, DSC, and NMR measurements, it was found that the presence of PAAm reduced deswelling and demixing in DN hydrogels compared to single-network (SN) ones. NMR spectroscopy showed decreased PNIPAm mobility due to agglomeration induced by the second network.

Li et al. developed strong antibacterial hydrogels using PAMPS and P(NIPAM-co-AAm) polymers [112]. The unique interpenetrating network structure enhanced mechanical strength (0.83–1.37 MPa) and thermo-sensitivity. In vitro evaluations showed biocompatibility, cytotoxicity, and antibacterial efficacy when loaded with a model drug. These DN hydrogels have potential for wound dressing and biomedical applications requiring strength around 1.0 MPa.

A combination of PNIPAM and PVA was chosen to enhance mechanical properties, particularly fracture toughness, of hydrogels [113]. PVA, known for forming nanocrystalline domains, offers the potential for DN alongside PNIPAm. The hydrogel was prepared through a multistep process involving modification of PVA with methacrylic acid (MA) and copolymerization with NIPAM using photoinitiation in a 3D-printed mold (Figure 6). The hydrogels exhibited high temperature responsiveness, toughness, and programmability, holding promise for soft robotics and actuator applications. DN hydrogels composed of PVA–borax as the first network and poly(NIPAM-co-AM) as the second network showed unique characteristics such as stretchability (up to 15 times to its original length), toughness (250% strain), thermo-responsive, resilience, and self-healing properties [114].

The thermal behavior of DN hydrogels was explored by incorporating PDEAm as a temperature-sensitive component alongside hydrophilic polymers such as PAAm, poly(*N,N*-dimethylacrylamide) (PDMAm), or PAMPS. This investigation involved swelling measurements, differential scanning calorimetry (DSC), and spectroscopic techniques [115,116]. The presence of a second hydrophilic network significantly affected their sensitivity to temperature. DN hydrogels exhibited less pronounced changes in deswelling, smaller enthalpy and entropy changes during phase transition, and a wider temperature range for the transition compared to single-network (SN) hydrogels. DN hydrogels retained more bound water due to interactions with the hydrophilic component. A novel thermodynamic model based on van’t Hoff analysis was developed to determine parameters such as enthalpy and entropy changes and critical temperatures and differentiating water states.

PDEAm/PAAm DN hydrogels were examined in terms of their responsiveness to three different stimuli [117]. Their response to temperature change, the presence and concentration of NaCl salt, and the composition of water–acetone was investigated. The hydrogel’s sensitivity to stimuli was influenced by introducing the second component and forming the double network. Due to the hydrophilic PAAm groups, temperature- and salt-induced changes during deswelling were less intense and gradual compared to those of the single-network hydrogel. The time-dependent deswelling and reswelling kinetics showed a two-step profile, corresponding to the release and absorption of solvent molecules during two processes with different characteristic times.

The concept of DN hydrogels led to the development of triple-network (TN) hydrogels as an advanced material design strategy, combining the benefits of multiple interpenetrating networks to achieve a balance of desirable properties for various high-performance applications [118]. PNIPAm was combined with alginate and conductive carbon nanofibers to create TN hydrogels [119]. These hydrogels achieved compressive strengths comparable to DN hydrogels but with additional benefits: thermal actuation from PNIPAm and electrical conductivity from the nanofibers. Potential applications for these TN hydrogels include soft electrodes for bionic or robotic devices, soft temperature or pressure sensors, and electronically controlled valves or liquid release systems. TN hydrogel, with excellent toughness, thermo-responsive, and self-healing properties, has been synthesized by copolymerizing NIPAm with PVA in the presence of tannic acid [120]. This hydrogel outperforms double-network hydrogels in mechanical strength. Its thermo-sensitivity slightly diminishes after a few cycles and then stabilizes. The hydrogel exhibits substantial self-healing with 81% efficiency, allowing it to repair itself, for example, when used as a valve in chemical reactors.

### 4.2. pH- and Temperature-Responsive Acrylamide-Based DN Hydrogels

Introducing DN structure for various combinations of hydrophilic and temperature-sensitive polymers is an effective approach to prepare pH- and temperature-responsive hydrogels. DN hydrogels were synthesized incorporating PNIPAm as a tightly cross-linked first network, polyacrylic acid (PAA) as a loosely cross-linked second network, and graphene oxide (GO) as an additive (Figure 7a) [121]. The impact of GO sheets and acrylic acid (AA) contents on various physical properties was investigated. The results indicate that PNIPAm/AA/GO hydrogels exhibit substantial volumetric changes in response to temperature and pH variations (Figure 7b,c), along with markedly rapid swelling/deswelling kinetics compared to conventional PNIPAM hydrogels. Furthermore, PNIPAm/AA/GO hydrogels demonstrate superior mechanical properties compared to their conventional hydrogels. Temperature- and pH-responsive DN hydrogels, akin to those formed with PNIPAm and PAA yet employing inorganic laponite as an effective cross-linker, exhibited remarkable swelling/deswelling behavior and mechanical strength [122].

Incorporating natural cross-linker genipin (GP) and water-soluble carboxymethyl chitosan (CMTC) in the preparation of DN hydrogels enabled the fabrication of biocompatible, mechanically robust dual pH- and temperature-responsive PNIPAm/clay/CMCTs/GP nanocomposite hydrogels [123]. Moreover, examining the drug absorption and release kinetics with acetylsalicylic acid (Aspirin) as the model drug indicated the potential for application as drug carriers. The hydrogels comprised a copolymer network of AM and sodium acrylate, complemented by an additional reversible network formed by PVA–borax complex, which exhibited exceptional swelling and stretching capabilities alongside pH sensitivity and self-healing properties [124]. Versatility was achieved through a dual cross-linking mechanism incorporating both physical and chemical bonds.

A novel bi-continuous double-network (BCDN) material was developed by Vancaeyzeele et al. [125]. The bi-continuous emulsion was employed to create poly(butyl acrylate) (PBA) and PAA amphiphilic material. This material was then swollen with a precursor of the second polymer PNIPAm, and after polymerization, PNIPAm and PAA formed DN within a bi-continuous templated material. This unique architecture offers several advantages, including enhanced mechanical properties from PBA and pH- and temperature-responsive qualities from the hydrophilic double network. The biomedical applications were explored, particularly for drug delivery with pH and temperature sensitivity. Furthermore, drug release from the amphiphilic bi-continuous structure varies with pH and temperature, allowing controlled release corresponding to physiological conditions.

## 5. Conclusions and Perspectives

The versatility of the responsive polymers, enabled by their tunable physical properties and responsiveness to multiple stimuli, underscores their potential in developing advanced materials for targeted applications such as smart drug delivery systems, tissue scaffolds, and actuators. IPN hydrogels offer significant advancements in this field, combining the properties of multiple polymer networks to enhance mechanical, thermal, and chemical characteristics. Their unique structure, formed by physically intertwined but not covalently bonded networks, allows for the integration of diverse functional properties, making IPNs highly adaptable to various applications. Acrylamide-based IPN hydrogels, in particular, have shown exceptional promise due to their enhanced mechanical properties and responsiveness to environmental stimuli. These hydrogels exhibit significant improvements in mechanical strength, stretchability, and stimuli responsiveness compared to single-component hydrogels. This is achieved by the synergistic combination of polymers with different properties, such as temperature sensitivity and pH responsiveness. Furthermore, the ability to fine-tune the phase transition temperature and responsiveness through the IPN strategy enables the creation of hydrogels tailored for specific applications. Innovations such as dual-temperature-responsive hydrogels and multi-stimuli-responsive IPNs expand the potential use of these materials in advanced biomedical and technological fields.

DN hydrogels have emerged as a significant class of IPN materials due to their exceptional mechanical properties, primarily their high fracture energy and toughness. Since the initial work by Gong et al. in 2003, DN hydrogels have been distinguished by their unique structure consisting of two interpenetrating networks with asymmetrical properties. This asymmetry, typically involving a tightly cross-linked first network and a loosely cross-linked second network, is crucial for their enhanced performance. Incorporating temperature-responsive polymers such as PNIPAm into DN hydrogels adds a layer of functionality, enabling applications in smart materials and biomedical devices. For example, DN hydrogels with PNIPAm have shown significant improvements in properties like deswelling rates and mechanical strength when combined with other materials such as polysiloxane nanoparticles or zwitterionic copolymers. Innovative synthesis techniques, including the use of natural cross-linkers and micropatterning, have further expanded the potential applications of DN hydrogels. These methods allow for the precise control of hydrogel properties, facilitating their use in advanced biomedical applications such as implantable biosensors and self-cleaning membranes.

Further refinement of IPN hydrogels can lead to more precise and controlled drug release mechanisms tailored for specific medical conditions. Integrating multi-responsive elements, such as in the form of TN hydrogels, could enable targeted drug delivery in response to specific physiological triggers, including pH, temperature, or the presence of specific biomolecules. Continued research into the fundamental interactions between different polymer networks within IPNs will enhance our understanding of how to design materials with specific properties. This knowledge will enable the creation of custom-tailored hydrogels for diverse applications. Ongoing research and development in the field of acrylamide-based IPN hydrogels are expected to further enhance their performance and expand their applicability, solidifying their role as a key material in the next generation of smart, responsive systems.

## Figures and Tables

**Figure 1 gels-10-00414-f001:**
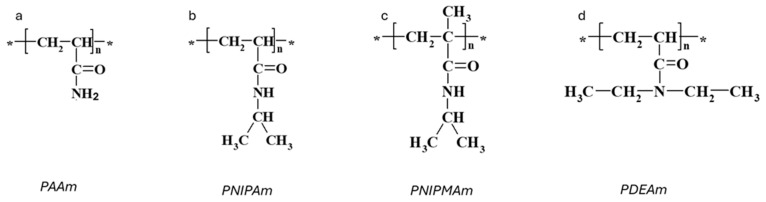
Chemical structures of responsive acrylamide-based polymers. Repeating units are marked with an asterisk *. (**a**) PAAm. (**b**) PINPAm. (**c**) PINPMAm. (**d**) PDEAm.

**Figure 2 gels-10-00414-f002:**
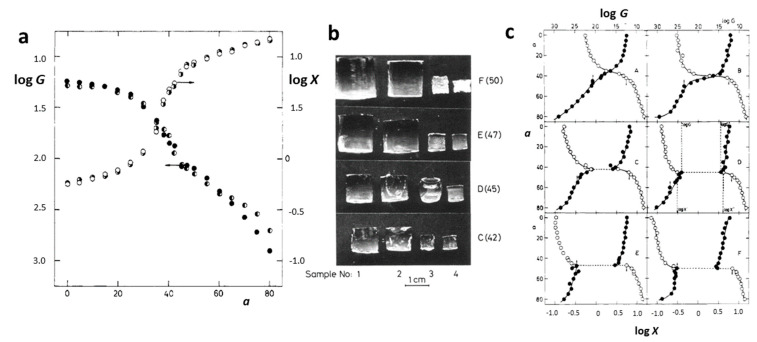
Dependence of the swelling ratio *X* and modulus *G* on the acetone content *a* for PAAm hydrogels (**a**). Sample dimensions determined at critical concentrations of the mixture water/acetone (**b**). Dependence of the swelling ratio and modulus on the acetone content for PAAm networks with increasing charge concentration (A, no charges; F, the highest charge concentration) (**c**). Reproduced with permission [27]. Copyright 1982, American Chemical Society.

**Figure 3 gels-10-00414-f003:**
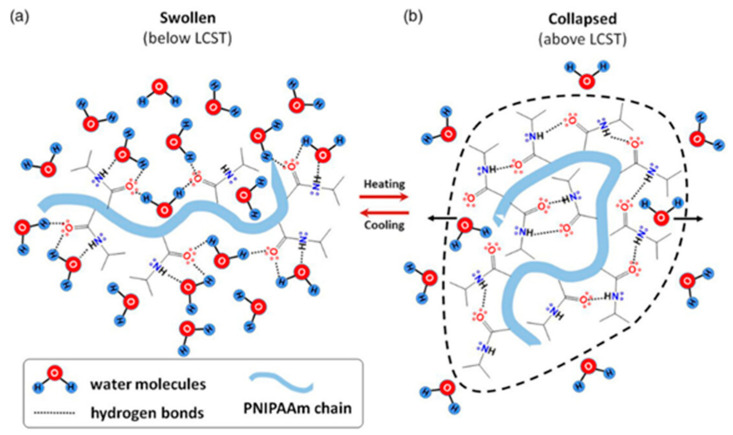
Schematic representation of the PNIPAm chains conformations and interactions: (**a**) swelling and (**b**) deswelling, triggered by temperature changes. Reprinted from [30], originally published under a CC BY 4.0 license.

**Figure 4 gels-10-00414-f004:**
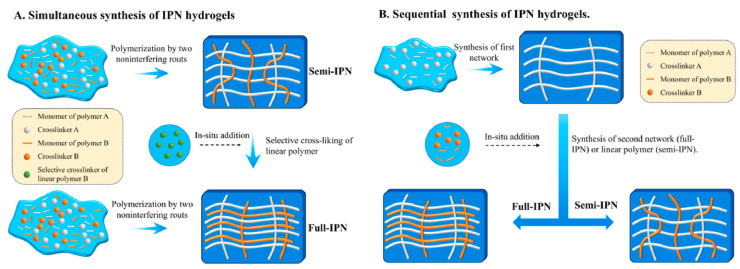
Methods of IPN synthesis. Reprinted from [63], originally published under a CC BY 4.0 license.

**Figure 5 gels-10-00414-f005:**
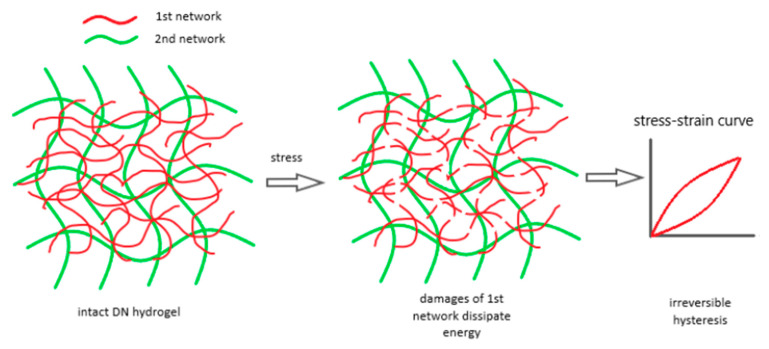
The damage process in DN hydrogels involves multiple breaks in the first network, which dissipate large amounts of energy and cause irreversible hysteresis in load–unload tests. Reprinted from [105], originally published under a CC BY 4.0 license.

**Figure 6 gels-10-00414-f006:**
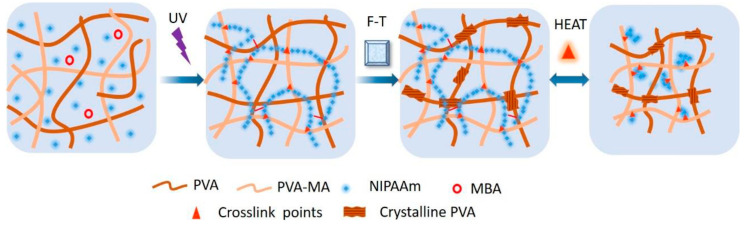
The forming and thermo-responsiveness mechanism of PVA/(PVA-MA)-g-PNIPAm hydrogels. Reprinted from [113], originally published under a CC BY 4.0 license.

**Figure 7 gels-10-00414-f007:**
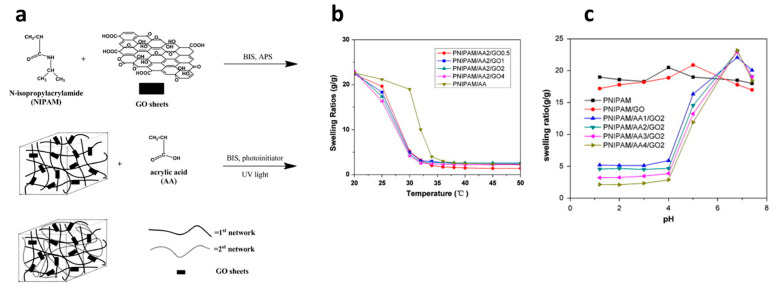
The synthetic procedure of pH- and temperature-responsive DN hydrogels PNIPAm/AA/GO (**a**) and their swelling ratios as a function of temperature (**b**) and pH (**c**). Reproduced with permission [121]. Copyright 2013, Elsevier B.V.

**Table 1 gels-10-00414-t001:** Overview of some relevant publications on IPN acrylamide-based responsive hydrogels.

IPN/SIPN Composition	Responsivity	Scientific Innovation	Refs.
SIPN PNIPAm/ether urethane-urea	Temperature	Release of heparin as a function of temperature and network composition	[80]
SIPN PNIPAm/hydrophilic polymer	Temperature	The impact of various hydrophilic polymers on the shift of the VPTT	[81]
IPN PNIPAm/Ca-alginate	Temperature	Hydrogel beads with thermally reversible core-shell structure for temperature-controlled drug release	[82]
		Porous hydrogels and model drug release	[72]
IPN PMAA/PNIPAm	Temperature, pH	Permeation studies on membranes for model drugs of varying sizes	[83]
IPN PVP/PAAm	Solvent	Ionic IPNs sensitive to solvent composition	[84]
SIPN PAAm/PNIPAm	Temperature	Detailed characterization of mechanical properties and their correlation swelling behavior	[85]
		Solubility of methylene blue and orange II for potential applications in temperature-controlled separation	[86]
SIPN PVA/P(NIPAAm-co-AAm), PVA/P(NIPAAm-co-AMPS)	Temperature	Influence of SIPN composition on temperature-induced transitions, swelling behavior, and mechanical properties	[87]
IPN PNIPAm/PNIPAm	Temperature	Improved intelligent characteristics, such as a controllable response rate based on the composition ratio of two network components	[88]
IPN PVA/P(NIPAAm-co-AAm), PVA/P(NIPAAm-co-AMPS)	Temperature	IPNs often do not shrink macroscopically, making them suitable for producing “gel-glasses”	[89]
SIPN PVA/ P(AAm-co-SMA)	Salt, pH	Salt and pH responsiveness for biomedical applications	[90]
IPN P(HEA-co-HEMA)/PNIPAm	Temperature	Potential to influence the rate of model drug release	[91]
SIPN P(VP-co-MAA)/PNIPAm	Temperature, pH	Porous hydrogels with fast pH- and temperature response as “on-off” switches in drug delivery	[92]
SIPN PUU/P(NIPAm-co-AA-co-BMA)	Temperature, pH	Dual-sensitive films for wound dressing	[93]
SIPN PDADMAC/PDEAm	Temperature, pH	Porous hydrogels with improved dual sensitivity	[56]
IPN PVA/PNIPAm	Temperature, pH	Modified PVA at different degrees of substitution	[94]
IPN PNIPAm/PASP	Temperature, pH	Improved shrinking and re-swelling	[77]
IPN PNIPMAm/PNIPAm	Temperature	IPNs formed with two temperature-sensitive components	[50]
SIPN salecan/PNIPAm	Temperature	Non-toxic hydrogels with enhanced cell adhesion	[95]
IPN PNIPAm/PVP	Temperature	Fully injectable hydrogels synthesized using innovative chemistry	[71]
IPN PNIPAm/PEG/Heparin	Temperature	Thermo-responsive and growth factor-affined hydrogels with highly adjustable mechanical properties	[96]
SIPN apo-GOx/PNIPAm	Glucose	Selective response to glucose concentrations of 0–20 mM with rapid, reversible volumetric changes; in vivo experiments	[97]
IPN PNIPAm/PNAGA	Temperature	Hydrogels with dual-temperature response, allowing simple-temperature monitoring and useful for electronic skin, wearable devices, bionics, and actuators	[73]

Abbreviations: methacrylic acid (MAA), 2-acrylamido-2-methylpropane sulfonic acid (AMPS), poly(1-vinyl-2-pyrrolidone) (PVP), poly(vinyl alcohol) (PVA), sodium methacrylate (SMA), 2-hydroxyethyl acrylate (HEA), 2-hydroxyethyl methacrylate (HEMA), N-vinylpyrrolidone (VP), polyurethane urea (PUU), acrylic acid (AA), butylmethacrylate (BMA), poly(diallyldimethylammonium chloride) (PDADMAC), polyaspartic acid (PASP), poly(ethylene glycol) (PEG), apo glucose oxidase (apo-Gox), and poly(N-acryloylglycinamide) (PNAGA).

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
