# Peer review of "Responsive Acrylamide-Based Hydrogels: Advances in Interpenetrating Polymer Structures"

_gels, 2024, doi:10.3390/gels10070414_

Round 1

Reviewer 1 Report

Comments and Suggestions for Authors

In this manuscript, the authors have summarized the review with emphasis on the functionality of acrylamide hydrogels composed of IPN or DN networks. Acrylamide-based hydrogels have been the subject of numerous studies, and it is very meaningful for the advancement of gel chemistry to summarize the latest research, as the authors have done in their manuscript as a review article.

The authors' manuscript is well structured, interesting, and suggestive, but contains the following minor issues that could be improved.

1)      Since this manuscript is a review, the authors should describe the differences in structure and properties between IPN and DN gels more clearly.

2)      The first network of DN gel should be selected from polymer chains with a characteristic chemical structure, but there is no description. The structural characteristics of the first network should be described in detail because they are essential for the mechanical properties of DN gels and the expression of specificity of DN gels.

3)      The authors summarize acrylamide-based IPN and DN gels, but is there any reason not to describe acrylamide-based triple network gels? If there are essential differences between the acrylamide-based IPNs and DNs and the acrylamide-based triple network, they should be described as good reasons not to address them in this review. Either way, the reviewer believes that the authors should address them in this review.

4)      This review does not deal with “surface properties of hydrogels,” which have been variously examined in acrylamide-based IPN and DN, but if there is a reason, it would be better to describe it.

5)      The figures could be more precise, and the authors should improve them.

Reviewer 2 Report

Comments and Suggestions for Authors

The manuscript is interesting and exhaustive, written in comprehensible way. I would suggest the authors to address the following points:

1. In Abstract and Introduction - please correct the name of Li to Lim. This is a fundamental work and must be cited correctly.

2. Please recheck chapter 2.1., more specifically if the PAAM from reference 26 is hydrolyzed. Probably you mean crosslinked.

3. A chapter concerning the in vivo properties of these gels will enhance the work.

Reviewer 3 Report

Comments and Suggestions for Authors

the manuscript is very interesting, it quickly exposes the characteristics advantages, disadvantages of interpenetrated hydrogels and the direction or trend of the research carried out with the different monomers derived of acrylamide.

I only see that there has been litle care preparing the figures, especially figures 2, 4, 6, and 7, the letters and nummers are very small, etc, that mus be improved,

I do support the publication of this manuscript after improvements.
